# Repeated ice streaming on the northwest Greenland continental shelf since the onset of the Middle Pleistocene Transition

Andrew M. W. Newton[1,2], Mads Huuse[1], Paul C. Knutz[3], and David R. Cox[1]

[1]Department of Earth and Environmental Sciences, University of Manchester, Oxford Road, UK, M13 9PL.

[2]School of Natural and Built Environment, Queen's University Belfast, University Road, UK, BT7 1NN.

[3]Department of Geophysics, Geological Survey of Denmark and Greenland, Øster Voldgade 10, 1350, Copenhagen, Denmark.

*Correspondence to*: Andrew M. W. Newton (amwnewton@gmail.com)

**Abstract.** Ice streams provide a fundamental control on ice sheet discharge and depositional patterns along glaciated margins. This paper investigates ancient ice streams by presenting the first 3D seismic geomorphological analysis of a major glacigenic succession offshore Greenland. In Melville Bugt, northwest Greenland, six sets of landforms (five buried and one on the seafloor) have been interpreted as mega-scale glacial lineations (MSGL) that provide evidence for extensive ice streams on outer palaeo-shelves. A gradual change in mean MSGL orientation and associated depocentres through time suggests that the palaeo-ice flow and sediment transport pathways migrated in response to the evolving submarine topography through each glacial-interglacial cycle. The stratigraphy and available chronology show that the MSGL are confined to separate stratigraphic units and were most likely formed during several glacial stages since the onset of the Middle Pleistocene Transition at ~1.3 Ma. The MSGL record in Melville Bugt suggests that since ~1.3 Ma, ice streams regularly advanced across the continental shelf during glacial stages. High-resolution buried 3D landform records such as these have not been previously observed anywhere on the Greenland continental shelf margin and provide a crucial benchmark for testing how accurately numerical models are able to recreate past configurations of the Greenland Ice Sheet.

## 1. Introduction

The northwest sector of the Greenland Ice Sheet (GrIS) is currently experiencing some of the largest mass losses across the ice sheet (Mouginot et al., 2019). During the Pleistocene the northwest sector has also been shown to have experienced major changes in ice margin extent through multiple glacial-interglacial cycles (Knutz et al., 2019). To better project the future evolution of the northwest Greenland ice sheet, and the GrIS as a whole, requires the reconstruction of past configurations of the ice sheet, the role and evolution through time of its ice streams, and an understanding of how the antecedent and evolving topography impacted ice flow patterns during past glacial stages. Typically, reconstruction involves using fragmented geological records to constrain or test numerical ice sheet models that attempt to map spatiotemporal changes in ice sheet extent and the dominant processes as the climate evolves across multiple glacial-interglacial cycles (Solgaard et al., 2011; Tan et al., 2018). Improving and building upon that fragmented geological record is, therefore, of considerable importance for helping to improve and calibrate these models – i.e. if models can accurately reconstruct the past, then we can have more confidence in what they project for the future.

Although much of the past offshore extent of the GrIS and its retreat is poorly resolved (Funder et al., 2011; Vasskog et al., 2015), there are some areas, such as the Uummannaq and Disko Troughs in the west and the Kangerlussuaq, Westwind, and Norske Troughs in the east and northeast of Greenland, that have been surveyed. Geophysical data and shallow marine cores have been used to document landforms from the Last Glacial Maximum (LGM) on the continental shelf, deglacial ages, and retreat styles – with retreat often punctuated by Younger Dryas stillstands and an intricate relationship between calving margins and ocean currents (Arndt et al., 2017; Dowdeswell et al., 2010; Hogan et al., 2016; Jennings et al., 2014; Sheldon et al., 2016). Seismic reflection data have been used to explore evidence of older glaciations and show that the GrIS repeatedly advanced and retreated across the continental shelves of west and east Greenland through much of the late Pliocene and Pleistocene (Hofmann et al., 2016; Knutz et al., 2019; Laberg et al., 2007; Pérez et al., 2018). These seismic data show that GrIS extent has varied by 100s km throughout the Pleistocene and offers additional constraining observations to borehole and outcrop data that provide conflicting evidence that Greenland could have been nearly ice-free or persistently ice-covered for parts of the Pleistocene (Bierman et al., 2016; Schaefer et al., 2016).

To help understand long-term climatic changes, especially those associated with ice streams during glacial maxima, landforms observed on palaeo-seafloor surfaces mapped from 3D seismic data can provide information on past ice

sheet geometries and ice streaming locations. Landforms can be observed on surfaces preserved within trough-
mouth fans (TMFs), typically deposited on the middle and upper continental slope, or on palaeo-shelf layers buried
on the middle and outer continental shelf that built out as the TMF prograded (Ó Cofaigh et al., 2003). Here, for
the first time offshore Greenland, buried glacial landforms preserved on palaeo-shelves are documented using 3D
seismic reflection data from Melville Bugt (Fig. 1). Whilst ice streams are thought to have been present in Melville
Bugt since ~2.7 Ma (Knutz et al., 2019), these landforms provide new, direct, and detailed evidence of ice flow
pathways for six episodes of ice stream advance onto the outer continental shelf of Melville Bugt since ~1.3 Ma.

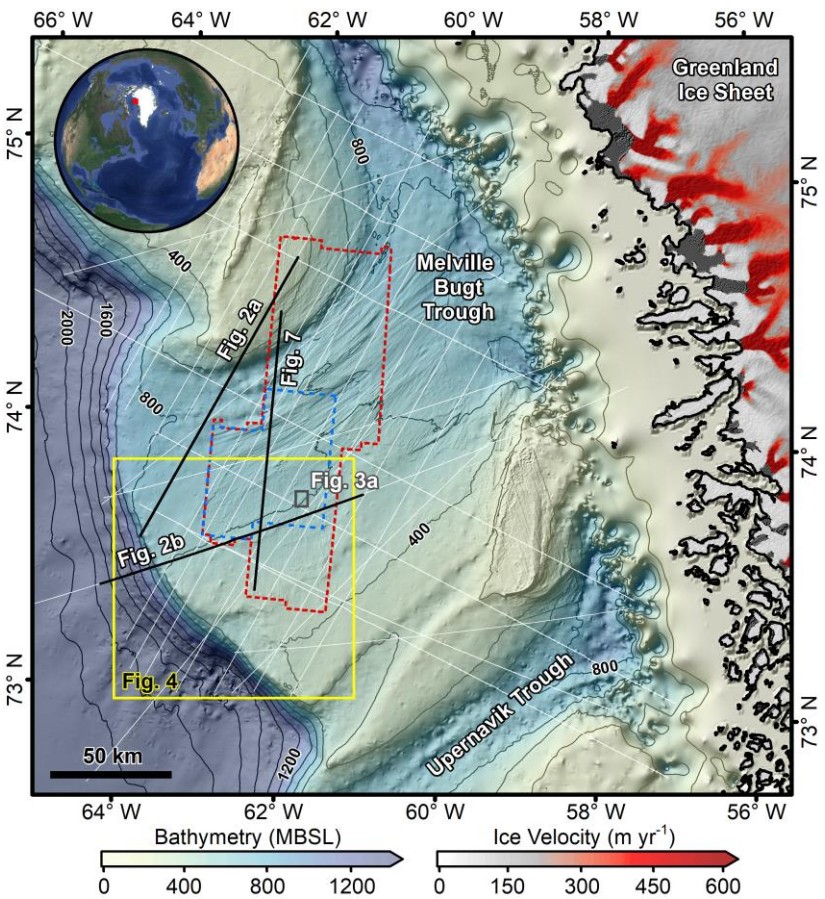


**Figure 1**: Seabed morphology and ice-flow velocity around the study area. The grey bathymetric contours are
every 200 m and the blue/red dashed lines show the outline of the 3D seismic surveys (blue is a high resolution
sub-crop of the original data that was reprocessed). The thin white lines show the locations of 2D seismic data.
Mean ice velocity from MEaSURES (cf. Joughin et al., 2010) shows contemporary outlet glaciers flowing into
northeastern Baffin Bay. Bathymetry combined from Jakobsson et al. (2012), Newton et al. (2017), and Knutz et
al. (2019). Locations of other figures shown. All figures plotted in UTM Zone 21N.

## 2. Background

Ice streams are corridors of fast-flowing ice that can measure >20 km wide and 100s km long, with velocities >400-500 m yr$^{-1}$ (Bennett, 2003). Both in the present and in the geological past, ice streams have been important conduits for ice sheet mass redistribution and sediment delivery to ice sheet margins (Vorren and Laberg, 1997). Mega-scale glacial lineations (MSGL) are elongated landforms (typically 1-10 km long) that form by the streamlining (Clark et al., 2003) or accretion of subglacial sediments (Spagnolo et al., 2016) beneath fast-flowing ice (Clark, 1993). This association is supported by observations of similar MSGL features beneath the present-day Rutford Ice Stream in West Antarctica (King et al., 2009). MSGL thought to date to the LGM have been observed on the present-day seafloor of the Melville Bugt study area (Fig. 1) and typically measure 4–6 km long, 100–200 m wide, and 10–20 m high (Newton et al., 2017; Slabon et al., 2016). The MSGL on the outermost continental shelf show that fast-flowing ice occupied the Melville Bugt Trough and reached the shelf edge, before retreating and experiencing changes in ice flow pathways, as is indicated by cross-cutting MSGL on the middle continental shelf (Newton et al., 2017).

The glacial stratigraphy in Melville Bugt (Fig. 1) extends across an area of ~50,000 km$^2$ and measures up to ~2 km thick. The succession records advance and retreat of the northwest GrIS across the continental shelf multiple times since ~2.7 Ma and is subdivided into 11 major prograding units separated by regional unconformities (Knutz et al., 2019). The stratigraphy is partly age-constrained by a number of dates extracted from microfossil (~2.7 Ma) and palaeomagnetic data (~1.8 Ma) (Christ et al., 2020; Knutz et al., 2019). These dates suggest that whilst sediment accumulation likely varied over orbital and sub-orbital timescales, over periods longer than this (0.5-1.0 Myr) it did not change substantially and was grossly linear through time since glacigenic deposition began (Knutz et al., 2019). In the northern part of the trough, topset preservation is limited due to more recent glacial erosion that has cut into the substrate (Fig. 2a), whereas in the south there is better preservation of aggradational topset strata (Fig. 2b) – i.e. palaeo-shelves where buried landforms might be found.

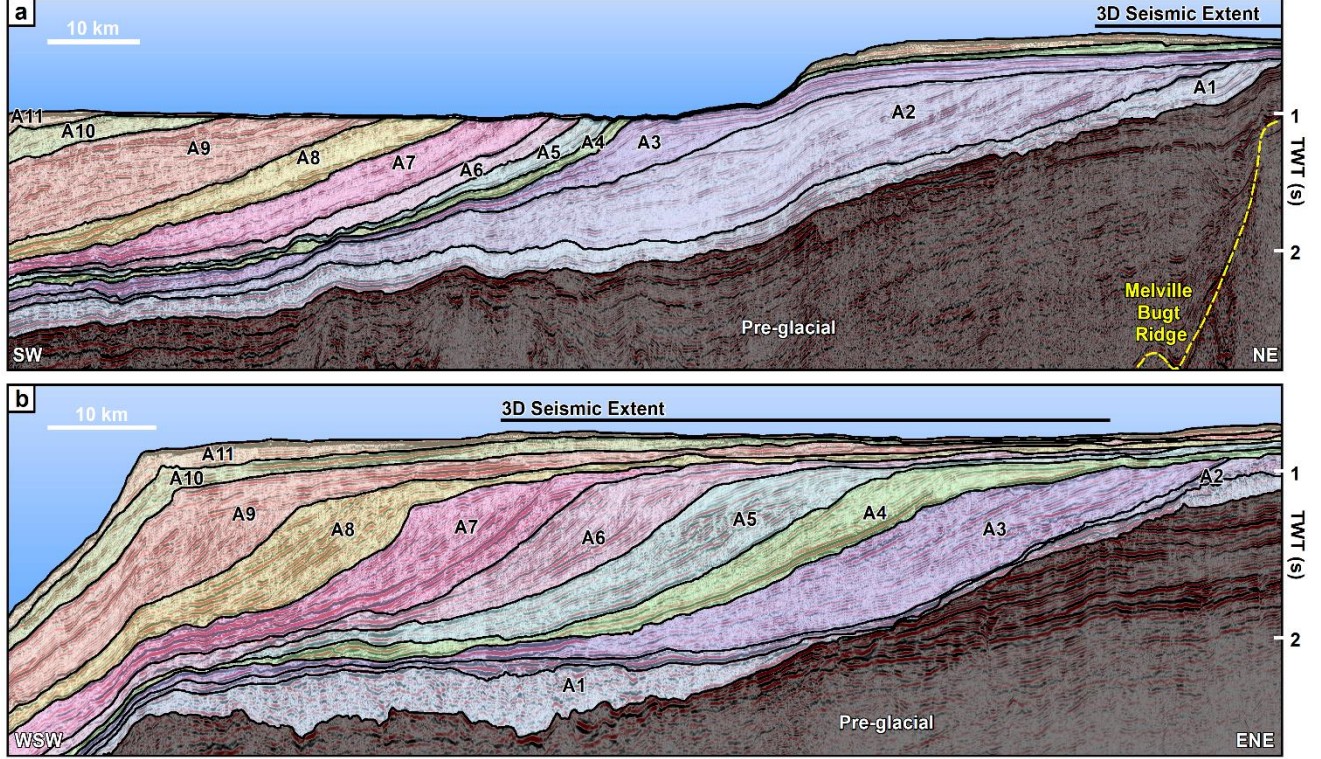

90

**Figure 2**: Seismic cross-section profiles through the glacigenic succession. The fan comprises 11 seismic stratigraphic units bounded by glacigenic unconformities formed since ~2.7 Ma (Knutz et al., 2019). The tentative chronology from Knutz et al. (2019) suggests that the palaeo-seafloor surfaces preserved within units A7-A9 likely cover a time period from ~1.3-0.43 Ma. This time period captures much of the Middle Pleistocene (781-126 ka) and the transition into it from ~1.3 Ma. Locations of the lines are shown on Fig. 1. TWT is two-way-travel time. Interpreted and uninterpreted seismic lines are provided as supplementary material.

## 3. Methods

This study used industry 3D and 2D seismic reflection data from Melville Bugt, northwest Greenland (Fig. 1). The vertical resolution of the glacial succession is ~10-15 m and the horizontal resolution ~20-30 m – based on frequencies ~30-50 Hz and a sound velocity ~2-2.2 km s$^{-1}$. Horizons were picked from within the 3D seismic data as part of a seismic geomorphological analysis (Posamentier, 2004), and gridded as 25x25 m two-way-travel time surface maps – i.e. buried palaeo-seafloor maps. It is important to note that unlike traditional seafloor studies carried out on bathymetric data, these palaeo-seafloor surfaces will have subsided and compacted since being buried. This means that landform thicknesses likely represent a minimum estimate of their original morphology. Seismic attributes, including variance and Root-Mean Square (RMS) amplitude, were extracted across the surfaces to aid

in visualising architectural elements and landforms. This study focused on identifying glacial landforms and used published examples to guide interpretation (e.g. Dowdeswell et al., 2016). Where possible, using the velocity model of Knutz et al. (2019), thickness maps were created for sub-units derived from deposits that were stratigraphically linked to surfaces containing glacigenic landforms – e.g. correlative slope deposits onlapping the profile of the glacially-influenced clinoform reflection. These depocentre maps can be used to document where sediments have been eroded and deposited, providing insight into how depositional patterns may have changed in response to the evolution of ice streams pathways. In the absence of precise dating for each surface, the linear age model of Knutz et al. (2019) has been used to relatively date glacial landforms identified in the different prograding units.

## 4. Subglacial landforms

Seismic geomorphological analysis of topset strata imaged in the 3D data showed four sets of buried streamlined features 5-15 km long and 200-300 m wide (Fig. 3 and 4). The landforms are typically 10-15 m high and although they are close to vertical seismic resolution limits (meaning that cross-sectional profiles are subtle) they are best observed in planform using the RMS amplitude or hillshaded surfaces. The streamlined features display a parallel concordance, are confined to individual palaeo-shelf layers within separate stratigraphic units, and their trend cross-cuts acquisition lines obliquely (Fig. 3 and 4). These features are interpreted as MSGL due to their morphology (Spagnolo et al., 2014), and similarity to MSGL observed on the local seafloor (Newton et al., 2017) and buried on other margins (e.g. Andreassen et al., 2007; Dowdeswell et al., 2006; Montelli et al., 2017; Rea et al., 2018).

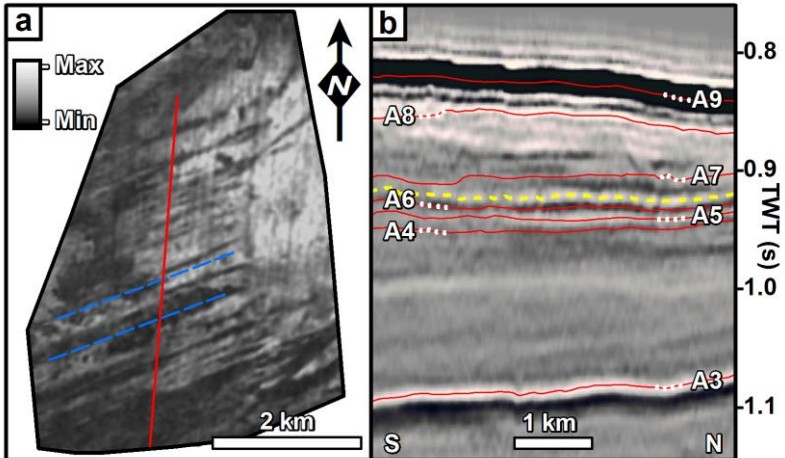

**Figure 3**: (**a**) MSGL set 1, the oldest example of mega-scale glacial lineations (blue dashed lines) displayed as an RMS image observed from 3D seismic reflection data and within unit A7 (b). The colour bar shows the maximum

and minimum RMS values. Note that this surface is only partially preserved due to subsequent glacial erosion. For location see Fig. 1. (**b**) Seismic cross-section profile showing the stratigraphic position (dashed yellow line) of the surface imaged in (a). The red lines show the top surface of each unit in the glacigenic succession and the dashed white lines are to help differentiate the labels to surfaces in this condensed stratigraphy. The location of the cross-section profile is shown by the red line on (a). Interpreted and uninterpreted seismic lines are provided as supplementary material.

MSGL set 1 is the oldest and is observed with an orientation of 254° on a partially-preserved surface in the lowest part of a condensed section of unit A7 (~1.3-1.05 Ma) (Fig. 3). It was not possible to confidently determine correlative slope deposits and the associated depocentre due to the limited spatial extent of their preservation. Rising through the stratigraphy, MSGL set 2 is observed in the upper part of unit A8 (~1.05-0.65 Ma) (Fig. 4a, d) and the associated depocentre is located in the southwestern part of the study area and measures up to 250 m thick. All of the sub-unit depocentres show sediment thicknesses greater than 100 m and have been mapped from the slope deposits that are correlative to the adjacent palaeo-shelves. The slope deposits are typically comprised of onlapping chaotic seismic packages interpreted as stacked glacigenic debrites (Fig. 5) (Vorren et al., 1989). MSGL set 2 has an average compass bearing of 225° ($\sigma = 5°$) that aligns well with the maximum depocentre thickness (Fig. 4a). MSGL sets 3 and 4 are observed on separate surfaces preserved within the topset strata of unit A9 (~0.65-0.45 Ma) (Fig. 4b, c, e, f,) and their bearings show a gradual transition to 237° from the 225° observed in unit A8 (Fig. 6).

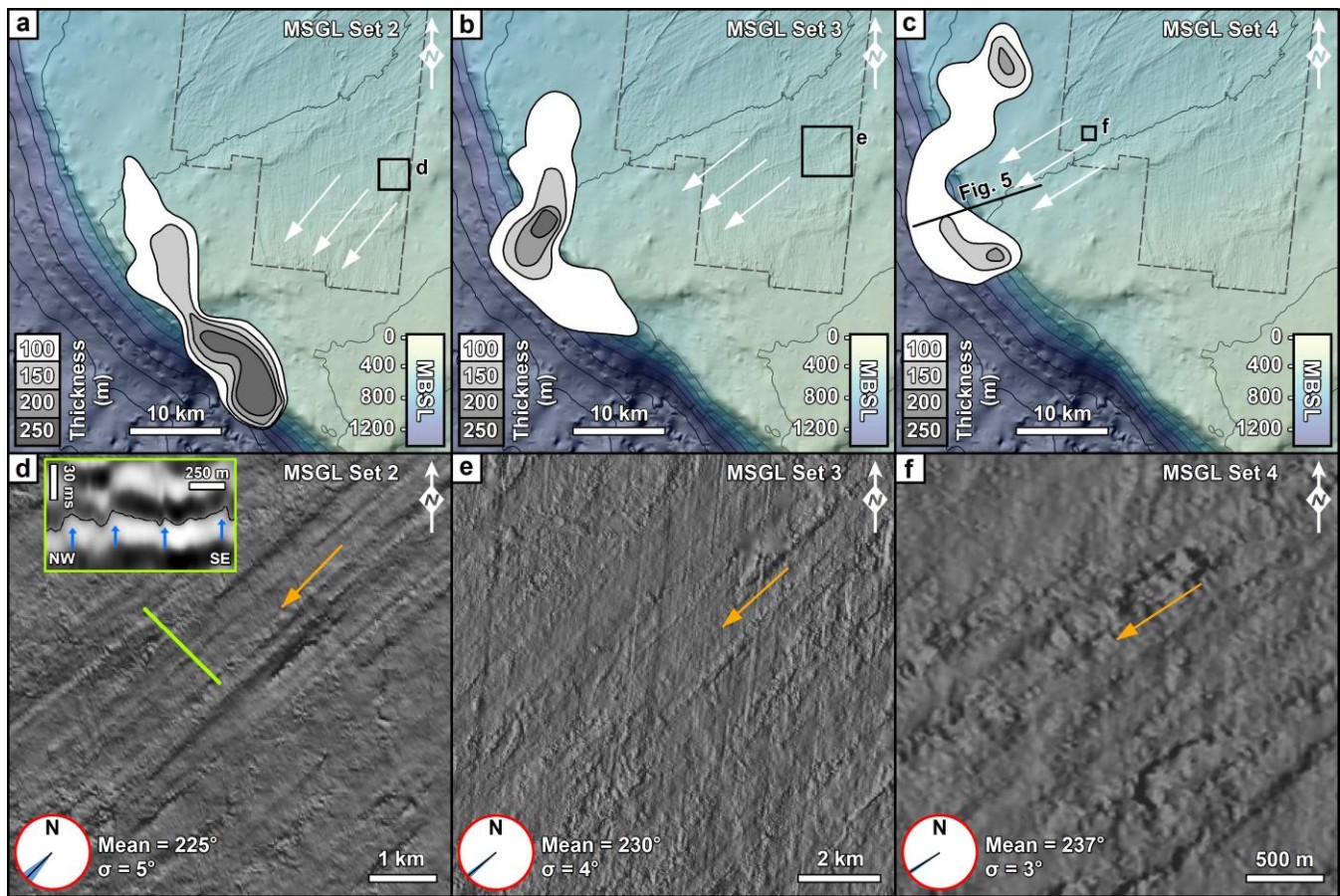

**Figure 4**: Buried MSGL and associated TMF thickness maps. Panels (**a**) to (**c**) show the geographic location of MSGL sets 2-4 which are displayed as hillshade images on panels (**d**) to (**f**). The dashed grey line on (a) to (c) is the 3D seismic survey outline overlain on the contemporary seafloor, the white arrows show the inferred ice flow direction from the MSGL, and the contoured outlines show the thickness of the sedimentary deposit associated with MSGL sets 2-4. Orange arrows on panels (d) to (f) show the inferred ice flow direction. On panel (d) the green line displays the location of the inset cross-section profile of the MSGL. Blue arrows point to the mounded features visible on the hillshade image. The red circles in (d) to (f) display average MSGL compass bearings (black line) and the standard deviation (blue fan beneath) for each panel. Location of panels (a) to (c) shown on Fig. 1. The relative ages and stratigraphic positons of each MSGL set are discussed in the text and labelled on Fig. 6.

Although the 3D seismic data do not cover the distal part of the succession, by using examples of MSGL that have been observed in 3D (Fig. 3, 4), the 2D seismic data were investigated for similar cross-sectional features. In unit A10 (~0.45-0.35 Ma) a reflection on the outer continental shelf shows a similar corrugated morphology, with heights of 10-15 m and widths of 200-300 m, to the MSGL pattern observed in the 3D data (Fig. 6b). The MSGL

documented in the 3D data also show that ice previously flowed towards this general area (Fig. 6c). The interpretation of the corrugated features as MSGL set 5 is less robust due to the lack of 3D data and whilst it is not possible to unequivocally rule out that these features are something else, such as iceberg scours, an interpretation of MSGL is supported by the location of these features in topset strata above the glacial unconformity that marks the top of unit A9, suggesting the presence of grounded and erosive ice on the outer continental shelf, conditions generally associated with MSGL formation.

The final set of MSGL (set 6) is observed in unit A11 (~0.35-0 Ma) on the seafloor and provides evidence for a grounded ice stream on the outer continental shelf at the LGM (Newton et al., 2017) (Fig. 6c). These MSGL show cross-cutting evidence that allow for changes in ice flow patterns to be deduced. The oldest MSGL on the seafloor suggest an ice flow towards the west-southwest that is parallel to the axis of the trough, whilst the younger MSGL (i.e. those which cross-cut the older MSGL) show an ice flow toward the south-southwest, suggesting a change in ice flow during deglaciation (Newton et al., 2017).

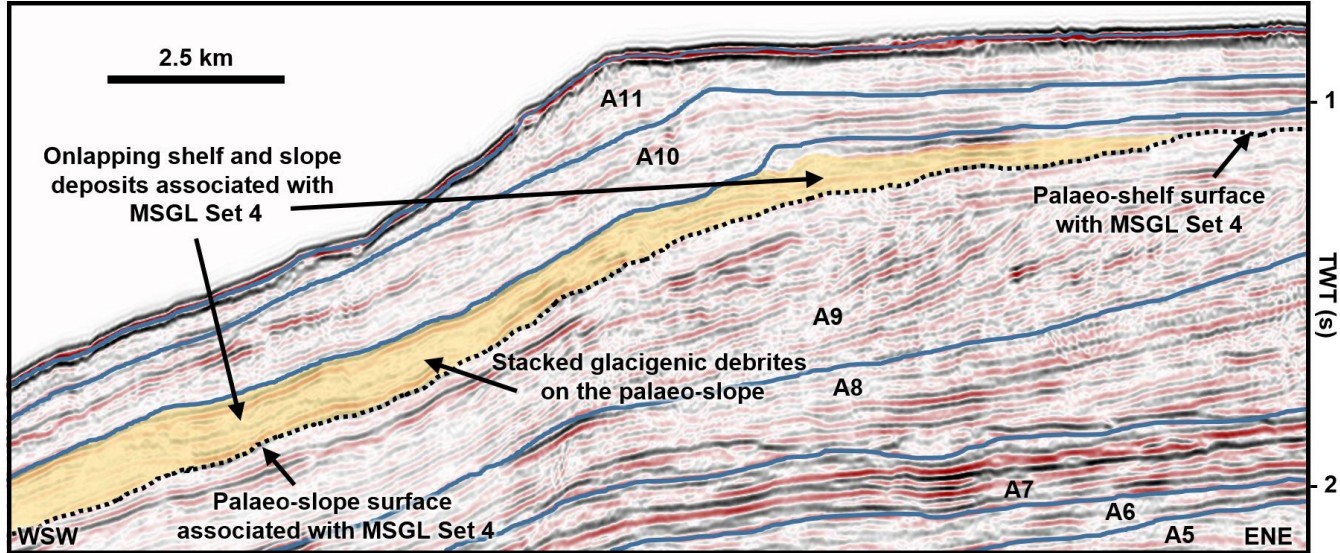

**Figure 5**: Seismic cross-section profile showing the main glacigenic units and the palaeo-shelf surface (dotted line) where MSGL set 4 is observed. Onlapping and stacked debrite packages are interpreted to be genetically linked to deposition caused by the ice stream that formed this set of MSGL and are used as an indicator of the broad depositional patterns displayed in Fig. 4c. Line location is shown on Fig. 4c. Interpreted and uninterpreted seismic lines are provided as supplementary material.

## 5. Palaeo-ice streams

The previous lack of 3D seismic data coverage means that prior to this study, ice stream landforms have not been observed for glacials preceding the LGM on the Greenland margin. Information on past ice flow patterns has, therefore, relied upon broad inferences from depocentre locations – i.e. areas where large volumes of sediment are associated with the general pathway of ice streams. Using the new seismic data, six sets of ice stream landforms have been documented – one on the seafloor, four buried surfaces imaged in 3D, and one captured in the 2D seismic. The MSGL sets provide evidence for multiple ice streaming events on the northwest Greenland continental shelf prior to, and including, the LGM. Limited chronological constraints are currently available to determine exact timings, but the available chronology suggests these features formed during six glacial stages after ~1.3 Ma (Knutz et al., 2019). Although no older MSGL have been imaged on palaeo-shelves captured in the available 3D seismic data, ice streams are inferred to have operated in the area prior to ~1.3 Ma, based on the large volumes of sediment delivered to the margin (Knutz et al., 2019). It is noteworthy that the first observations of MSGL occur at the onset of a major change in the depositional patterns of the Melville Bugt and Upernavik TMFs. Unit A7 was deposited when the Melville Bugt and Upernavik TMFs combined to form an elongate depocentre up to 1 km thick. During the subsequent deposition of unit A8 the TMFs separated into discrete depocentres up to 700 m thick, signalling a possible reorganisation in ice flow in the region (Knutz et al., 2019). The reasons for this change are unresolved, but modification of the submarine topography brought about by glacigenic deposition and erosion, such as presented here, may have forced adjustments in the ice sheet flow on the outer continental shelf due to changes in available accommodation.

Switches in ice stream pathways on continental shelves between different glacial maxima have been observed on the mid-Norwegian margin, where new cross-shelf troughs were formed through the erosive action of ice (Dowdeswell et al., 2006). In contrast to the mid-Norwegian margin, the available data in Melville Bugt does not show evidence of buried cross-shelf troughs. The observations show changes in ice stream pathways that appear to have occurred more gradually between each MSGL set but remained focused within the confines of the pre-existing trough. The longevity of the Northern Bank and the significant overdeepening of the inner trough (cf. Newton et al., 2017) likely provided consistent topographic steering of ice streams on the inner continental shelf. On the outer continental shelf, deposition during the preceding glacial stage likely forced gradual ice stream migration northward due to this deposition reducing the available accommodation for subsequent glacial stages (Fig. 7). Thickness maps associated with MSGL sets 2-4 demonstrate this gradual, rather than extreme, shift in ice stream drainage pathways

that is supported by 5-6° shifts in the mean orientation of each MSGL set from 225° during unit A8 time, to 237°
during unit A9 (Fig. 4). This shift continued at the LGM where the majority of MSGL on the outer continental
shelf – except for some cross-cutting related to deglaciation (Newton et al., 2017) – show a mean orientation of
~248° (Fig. 6c).
The partial preservation of the different palaeo-shelves means ice margin fanning on the less topographically-
confined outer continental shelf cannot be definitively ruled out as an explanation for differing MSGL orientations.
However, the observed metrics and depocentre migration provide complementary evidence that this was in
response to a gradual migration of the main ice stream flow pathway – i.e. ice flow pathways gradually moved
northward in a clockwise pattern from unit A8 onwards (~1 Ma). The gradual shift northward of the main ice stream
pathway and its associated erosion meant that topset deposits in the south, with each passing glacial stage, were
increasingly less impacted by the ice stream erosion and therefore the landforms that they contained had a better
chance of being preserved through subsequent glacial stages. The Melville Bugt Trough is the widest in Greenland
(Newton et al., 2017) and it is possible that the preservation of these topsets is a consequence of this. The
preservation suggests that whilst the main palaeo-ice stream trunks associated with each glacial stage were
accommodated within the broad confines of the trough, the fast-flowing and most erosive ice did not occupy its
full width – e.g. there are no MSGL present for the LGM (set 6) in the southern part of the trough. The northward
migration of the main ice stream pathway is also reflected by erosion and cutting into the deposits of the Northern
Bank (Fig. 7). Although ice stream margin fanning or changes in upstream ice sheet controls cannot be ruled out,
the gradual depocentre and MSGL migration suggests that deposition during successive glacial stages may have
been sufficient to bring about small changes in flow directions and subsequent depositional patterns. Future ice
sheet modelling can contribute to this discussion by exploring whether ice volume over northern Greenland would
have been sufficient to maintain ice flux if the ice streams occupied the full width of the Melville Bugt Trough. To
a lesser extent, it is possible that the Melville Bugt Ridge, an underlying tectonic structure which has previously
generated accommodation in the southern part of the basin through differential subsidence (Cox et al., 2020; Knutz
et al., 2019), could have contributed to reducing potential erosion of aggradational topsets by increasing palaeo-
water depths to the point where ice grounding was significantly reduced or removed.

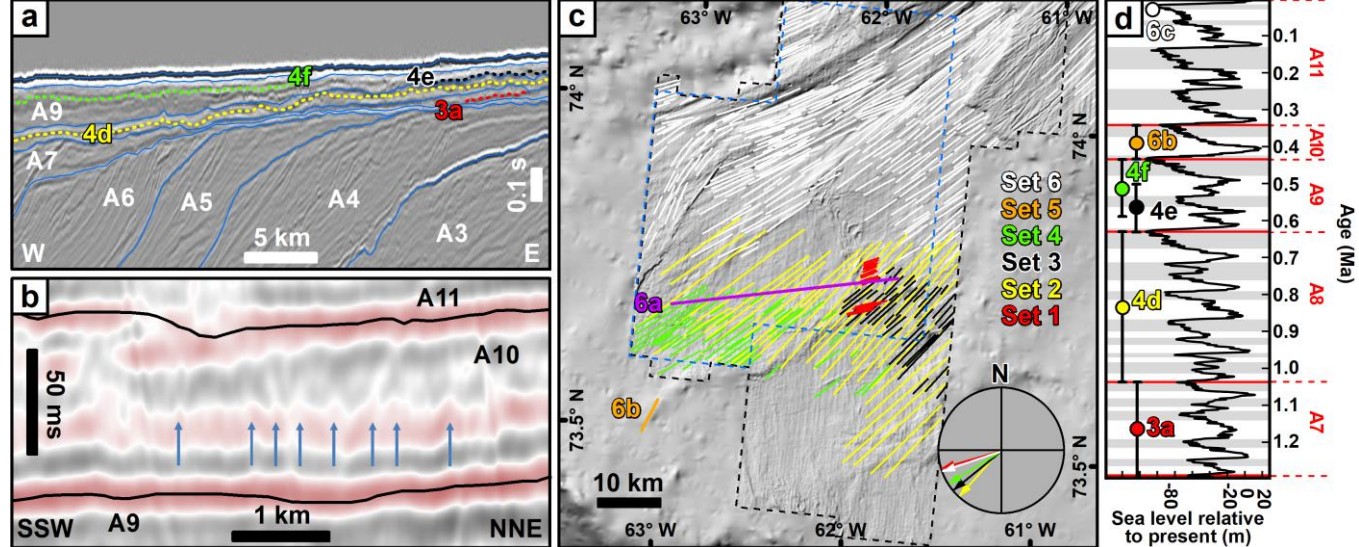

**Figure 6**: (**a**) Seismic cross-section profile showing the stratigraphic location of the surfaces shown in Fig. 3 and 4. The blue lines are the tops of the units shown on Fig. 2. The location of the line is shown on Fig. 6c. (**b**) Seismic cross-section profile from 2D seismic survey showing evidence for potential MSGL (blue arrows) in unit A10 on the outer continental shelf. Seismic line location is shown on Fig. 6c. (**c**) Digitized MSGL record from 3D seismic data. Set 6 represents the LGM record from Newton et al. (2017) and sets 1-5 from the current study. The compass shows the mean bearings for each set of MSGL with the exception of set 5 because it is not captured in 3D. (**d**) Possible age range for each MSGL surface observed within the glacigenic units of Knutz et al. (2019) and compared against the global sea level record (Miller et al., 2011). Grey bands are glacial stages. Note that in all the panels, the surfaces (a), digitised MSGL (c), mean flow bearings (c), and labels (d) are colour-coded to ease cross-referencing. Interpreted and uninterpreted seismic lines are provided as supplementary material.

In the wider context of the whole GrIS, in east Greenland, sedimentological and geophysical evidence suggest that early in the Middle Pleistocene Transition (MPT - ~1.3 Ma to 0.7 Ma) ice advanced across the continental shelf (Laberg et al., 2018; Pérez et al., 2019), whilst offshore southern Greenland documentation of increased ice-rafted detritus suggests a similar ice advance (St. John and Krissek, 2002). MPT ice sheet expansions have been documented in the Barents Sea (Mattingsdal et al., 2014), on the mid-Norwegian margin (Newton and Huuse, 2017), the North Sea (Rea et al., 2018), and in North America (Balco and Rovey, 2010), highlighting a response of all major Northern Hemisphere ice sheets to a currently unresolved climate forcing. Although ice streaming in Melville Bugt continued after the MPT and through to the latest Pleistocene, some studies from lower latitude areas

of west and east Greenland show reduced ice stream erosion and deposition at this time (Hofmann et al., 2016;
Pérez et al., 2018), perhaps suggesting the high latitude locality of Melville Bugt or the overdeepened and
bottlenecked geometry (topographic constraints) of the inner trough (Newton et al., 2017) helped promote
conditions favourable for ice streaming.

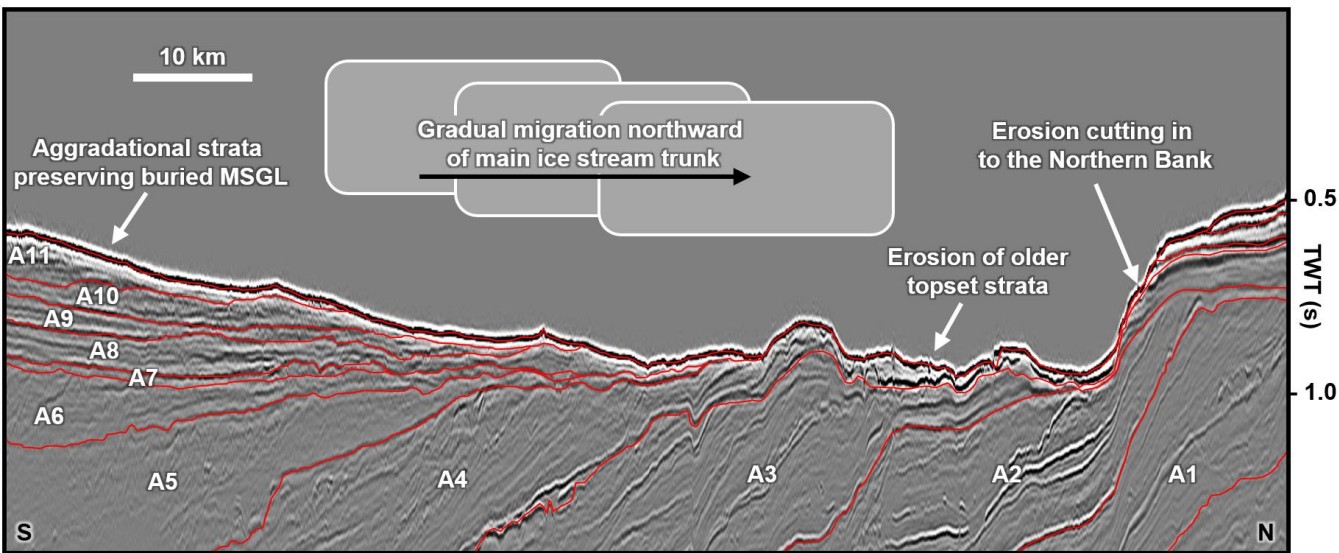

**Figure 7**: Interpreted seismic strike cross-section profile across the continental shelf showing spatially variable
preservation of topset deposits associated with the main depositional units. This variable preservation is thought to
relate to the gradual migration of the ice stream away from the areas of higher topography that contain the
aggradational strata. This northward migration of the ice stream pathways is also reflected by the erosion of the
southern flank of the Northern Bank. Location of the line is shown on Fig. 1. Interpreted and uninterpreted seismic
lines are provided as supplementary material.

The MSGL record presented here provides some additional insight into the contradictory records on the longevity
of the GrIS. Schaefer et al. (2016) showed that cosmogenic signatures require ice-free periods during the
Pleistocene and whilst these ice-free periods need not have occurred since 1.1 Ma, ice sheet loss could have
occurred during or after the MPT. Ice stream evolution has been shown to have led to rapid ice sheet changes on
other ancient ice sheets (Sejrup et al., 2016), and given that ~16% of the GrIS currently drains into Melville Bugt
(Rignot and Mouginot, 2012) the ice streams documented here could have contributed to major changes in ice sheet
organisation and extent – indeed, the numerical model used by Schaefer et al. (2016) requires the early loss of the
northwest GrIS during ice sheet collapse. Fully resolving issues like this requires numerical ice sheet models that
are capable of reproducing fragmented geological evidence. For example, recent modelling exploring Pleistocene
climate evolution (Willeit et al., 2019) provides palaeo-geographic maps of ice sheet extent that do not capture the
ice sheet extent inferred from buried landform records on many glaciated margins (e.g. Rea et al., 2018), including
Melville Bugt. Thus, there is currently a mismatch between modelling outputs and landform records. If these
models are not able to recreate ice sheet extent, ice stream locations, and flow pathways that have been extracted
from the geological record then those models will require refinement before they can be used as a tool for projecting
future GrIS evolution. These potential discrepancies underline how geological records, such as those presented
here, provide crucial empirical constraints for modelling the GrIS across multiple glacial-interglacial cycles.

**6. Conclusions**
This study provides a seismic geomorphological analysis offshore northwest Greenland and documents, for the
first time, several sets of buried MSGL on the Greenland margin. The observation of different MSGL sets in
separate stratigraphic layers confirms the presence of fast-flowing ice streams during at least six glacial maxima
since the onset of the Middle Pleistocene Transition at ~1.3 Ma. These landform records show that grounded and
fast-flowing ice advanced across the continental shelf to the palaeo-shelf edge of northwest Greenland, with each
subsequent ice stream flow pathway being partly controlled by the deposits left behind by the ice streams that
preceded it. This represents a first spatio-temporal insight into sediment deposition and ice flow dynamics of
individual ice streams during glacial maxima since ~1.3 Ma in Melville Bugt. These results help to further
emphasise why northwest Greenland would be suitable for future ocean drilling that will help to elucidate ice sheet
and climate history of the region.

**Data availability**
The Geological Survey of Denmark and Greenland or the authors should be contacted to discuss access to the raw
seismic reflection data.

**Author contribution**
AMWN carried out the seismic geomorphological study, drafted the figures, and wrote the initial text. All other
authors contributed to interpretation and manuscript preparation.

**Competing interests**
There are no competing interests to declare.

**Acknowledgements**
AMWN was supported by the Natural Environment Research Council (NERC - NE/K500859/1) and Cairn Energy.
DRC was funded by NERC and the British Geological Survey (NE/M00578X/1). Schlumberger and ESRI are
thanked for Petrel and ArcGIS software. All authors thank Cairn Energy and Shell for data and permission to
publish. Simon H. Brocklehurst is thanked for pre-reviewing this work and offering valuable insights. Brice R.
Rea, Lara F. Perez, an anonymous reviewer, and the editor Pippa Whitehouse are thanked for helpful comments
and handling of the manuscript.

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
