# Peer review of "shelf since the onset of the Middle Pleistocene Transition"

_The Cryosphere, 2019_

## Referee Comment (RC1) · Lara Perez (Referee) · 2 Jan 2020

Dear Editor and Authors, It has been a pleasure review your manuscript 'Repeated ice streaming on the northwest Greenland shelf since the onset of the Middle Pleistocene Transition'. I find the manuscript in a very good shape and ready for publication after minor revisions. This manuscript constitutes an important contribution to our understanding of glacial-related systems. In addition, future ice sheet models can take advantage of the insights provided here. My main concern is regarding the consideration of the seismic horizon gridded maps as palaeo-seafloor maps. Even though, we make this extrapolation often, it should be mentioned in the manuscript that the

maps of the palaeo-surfaces presented have not been backstripped or decompacted. Therefore, variations with respect the original morphology of the palaeo-seafloor are expected. In addition to this, I would appreciate to see the seismic profiles and maps without the overlapped interpretation (e.g. Fig. 2, 3, 5 and 6). Perhaps the same sections of the profiles can be included as supplementary material that the reader can check if needed. Finally, I have a few minor suggestions that can perhaps contribute to the improvement of the manuscript. Line 12: They are actually 6 sets of landforms considering the ones of the seafloor previously described. I suggest to rephrase this sentence to include them all. Line 30: Here and elsewhere you include important information in brackets. I suggest to limit the brackets and include these statements within the main text. Line 33: I know there are many examples, but could you give a couple of main references in case the reader wants to check other works? Lines 35 to 41: This sentence is long and difficult. Could you split the information here? Line 55: How this fit with Knutz et al., 2019 and the ice advance at aprox. 2.3 Ma? Lines 71 to 73: Perhaps this part fits better in the Introduction section feeding the discussion regarding the lack of previous evidences. Here you can develop further (2-3 sentences) the description of the seafloor MSGL which can become more important in the discussion regarding the change in time of the MSGL patterns. Line 77: Knutz et al., 2019. There are any previous references on this? Line 96: Please clarify that the surface maps have not been backstripped or decompacted and these processes can have an important impact in the original morphology, particularly on the deepest sections. Lines 102 to 103: This is close to interpretation. Line 105: It would be very interesting to have a more accurate age model. Perhaps, the future drilling proposals help on this. Lines 113 to 114: This is also interpretation. Can be somehow moved to the discussion, so you keep here plain description? Line 115: e.g. There are more works focus on MSGL. Line 165: Newton et al. A short description of the seafloor MSGL should be included in this work too. Line 193: cannot not? Line 196: (1 Ma) onwards. Add the age information.

Congratulations for the well-done work. Best Regards, Lara F. Perez

---

## Referee Comment (RC2) · Anonymous Referee #2 · 15 Apr 2020

It has been a pleasure to read this very well written manuscript. I am not a geophysicist, but I find the presentation of the geophysical data and the interpretations of the data to be very logical and understandable. The authors do a good job of outlining why this unusual record of multiple periods of ice streaming to the palaeo shelf edge is important and what its implications are.

I have only one complaint. Given how unusual it is to have preservation of aggradational topset strata and given the paucity of 3-D seismic data around Greenland (I presume it is not abundant), how do we know how extensive ice streams were elsewhere? In the abstract, but not in the conclusions, you state: 'This suggests that the ice

streams that occupied Melville 21 Bugt during the Middle and Late Pleistocene were more active and extensive than elsewhere in Greenland.' I think it is less presumptive and likely more correct to state that these have not been observed elsewhere around Greenland. If other areas of the palaeo shelf had 3-D seismics and aggradational topset strata, then perhaps one would find that Melville Bugt is not the only area where ice advanced multiple times to the palaeo shelf edge. As you state, the record of past glacier ice extent is fragmentary. You have found a very informative fragment and even better, you can correlate your unusual preserved shelf strata to slope deposits (Knutz. Et al., 2019). I think it is more likely than not that many parts of the Greenland shelf had extensive ice at the same times as shown by the Melville Bugt record. Modeling might actually show what type of configuration of the ice sheet is likely and where/when more of these deposits might be found, guided by your observations. I think the conclusions are well written and would be a good model for revising the abstract.

The unusual preservation of topset beds is interesting. Why is there is accommodation space in Melville Bugt. You say it is because the ice streams are not occupying the whole trough and so not eroding previous sediments. Did I understand that correctly. Is Melville Bugt unusually wide compared to other troughs? It looks like it is wider than most. A couple of minor comments: P9 line 166…delete repeated … and say multiple steaming events. 193…cannot not. Double negative or needs editing? 199 and another example Line 241 and other places in the manuscript. Unattended this. Try to add what 'this' is each time, just as you did in line 196, where you say 'this gradual shift'. It will make the writing even more clear.

---

## Author Comment (AC1) · 16 Apr 2020

The authors would like to thank Dr Lara Perez for insightful comments that will help improve the readability and robustness of the paper. The issue of compaction and subsidence is important and we will add that narrative in as suggested. We are happy to make all the suggested text edits. One small point regarding the comment about line 113-114: this does cross into the lines of interpretation as pointed out, but this was intentional as we wanted to present this section as a description and interpretation section as we felt it would help the flow of what is a relatively short paper. So we hope to keep this as is, but with a slight rewording to improve the flow of the text, happy to

discuss if a change is deemed necessary. A supplementary document will be created showing the seismic profiles with and without interpretations overlain as suggested. We think that all of Dr Perez's comments can be addressed relatively quickly and simply. Thank you for taking the time to review our manuscript.
* * *

---

## Author Comment (AC2) · 16 Apr 2020

The authors would like to thank the reviewer for their comments on how to improve the readability and robustness of the paper. All the minor text changes will be made as suggested. The main complaint is a reasonable one and is actually in line with our own thinking. This requires a small amount of wordsmithing, as the reviewer has suggested, and we are happy to do that. The short narrative on the available accommodation will also be tweaked to make our thinking clearer, in response to the reviewer comment. We are confident that all of the comments can be addressed relatively quickly and simply. Thank you for reviewing our manuscript.

---

## Author Response (AR1)

[revised manuscript text omitted]

**Response to reviewers**

The authors are grateful to the editor and reviewers for their helpful comments. All comments have been acted upon and the changes made as suggested.

Editor – Dr. Pippa Whitehouse

I would like to thank both reviewers for their constructive comments on this manuscript and also the authors for posting their response to the reviewers' comments.

In their response, the authors outline the steps that they propose to take to address all the main points raised by the reviewers. I request that the authors consider two additional points as they prepare a revised version of their manuscript (the first is taken from my original access review):

- The authors partly motivate their study by referring to ongoing discussions about the timing and extent of ice sheet minima during the Pleistocene. However, given that this study presents evidence relating to ice sheet maxima, the link is a little weak. The desire to provide more-complete geological evidence to test ice-sheet model reconstructions is well posed, but you may want to consider whether being able to reproduce ice sheet behaviour during a glacial period automatically qualifies an ice sheet model as being suitable for projecting future Greenland Ice Sheet change.

  Response: The editor has raised a fair point about this linkage and we have modified the text to make the link clearer. Our thinking behind this justification is that these snapshots provide a test for how accurately numerical ice sheet models recreate the past. If they are able to reproduce the observations for glacial maxima (such as ours), then we can, perhaps, have more confidence that the underlying physics contained within these models is a good representation of reality. Therefore, when these models are used to explore glacial minima we can have more faith in their outputs. An additional point, which is in the discussion section, relates to how ice streams at glacial maxima can have a significant bearing on how ice sheets retreat during the transition to minima (the role of ice streams in the North Sea being a good example). So we believe the link is justifiable and have made the text in the opening paragraph clearer to reflect this.

- Please ensure that, where necessary, statements are suitably supported by references.

  Response: All locations suggested by the reviewers that would benefit from additional references have had them added.

In general, both reviews are positive, and I therefore encourage you to submit a revised manuscript that addresses the points mentioned above and in the individual reviews.

Kind regards,

Pippa Whitehouse

 Reviewer #1 – Dr. Lara Perez

Dear Editor and Authors, It has been a pleasure review your manuscript 'Repeated ice streaming on the
northwest Greenland shelf since the onset of the Middle Pleistocene Transition'. I find the manuscript in a very
good shape and ready for publication after minor revisions. This manuscript constitutes an important
contribution to our understanding of glacial-related systems. In addition, future ice sheet models can take
advantage of the insights provided here.

Response: The authors would like to thank Dr. Lara Perez for her insights and helpful comments that will
improve the readability of our manuscript and its robustness. We also thankful to Dr. Perez for the quick
turnaround of the review.

My main concern is regarding the consideration of the seismic horizon gridded maps as palaeo-seafloor maps.
Even though, we make this extrapolation often, it should be mentioned in the manuscript that the maps of the
palaeo-surfaces presented have not been backstripped or decompacted. Therefore, variations with respect the
original morphology of the palaeo-seafloor are expected.

Response: Important point and we have added this narrative in as suggested.

In addition to this, I would appreciate to see the seismic profiles and maps without the overlapped
interpretation (e.g. Fig. 2, 3, 5 and 6). Perhaps the same sections of the profiles can be included as
supplementary material that the reader can check if needed.

Response: Very happy to include supplementary material with the revised version showing the un-interpreted
and interpreted seismic profiles.

Finally, I have a few minor suggestions that can perhaps contribute to the improvement of the manuscript.

Line 12: They are actually 6 sets of landforms considering the ones of the seafloor previously described. I
suggest to rephrase this sentence to include them all.

Response: Changed as suggested.

Line 30: Here and elsewhere you include important information in brackets. I suggest to limit the brackets and
include these statements within the main text.

Response: Changed as suggested.

Line 33: I know there are many examples, but could you give a couple of main references in case the reader
wants to check other works?

Response: A couple of references referring to the issues of piecemeal geological/glacial reconstruction have been added.

Lines 35 to 41: This sentence is long and difficult. Could you split the information here?

Response: Agreed. Changed as suggested.

Line55: How this fit with Knutz et al., 2019 and the ice advance at aprox. 2.3 Ma?

Response: Important point. These new results do not suggest ice advance from Knutz et al. (2019) is wrong, but instead build upon that by adding confidence to some of the interpretations by providing direct and definitive evidence of ice stream landforms. Text has been clarified to make sure this is clear.

Lines 71 to 73: Perhaps this part fits better in the Introduction section feeding the discussion regarding the lack of previous evidences. Here you can develop further (2-3 sentences) the description of the seafloor MSGL which can become more important in the discussion regarding the change in time of the MSGL patterns.

Response: Good suggestion. Implemented as suggested with some extra description of the seafloor MSGL.

Line 77: Knutz et al., 2019. There are any previous references on this?

Response: Reference to key study added in as suggested.

Line 96: Please clarify that the surface maps have not been backstripped or decompacted and these processes can have an important impact in the original morphology, particularly on the deepest sections.

Response: Clarified as suggested.

Lines 102 to 103: This is close to interpretation.

Response: Edited to make this clearer.

Line 105: It would be very interesting to have a more accurate age model. Perhaps, the future drilling proposals help on this.

Response: Agreed that the age model is a key issue and is something we explored refining in a number of ways
by trying to sync the landform records with other proximal ODP/IODP records with good dating chronologies.
However, whilst this is potentially insightful it is not robust enough to refine the age model confidently so we
elected not to include this (also based on comments at AGU from colleagues that raised similar discussions).
Not sure if a text edit was required or if this was just a general comment.

Lines 113 to 114: This is also interpretation. Can be somehow moved to the discussion, so you keep here plain
description?

Response: Fair point. Though we have written this section as a Description-Interpretation section as that helps
prevent repetition of how descriptions inform the interpretation, which we thought was preferable for a short
paper like this. We would prefer to keep it as is, but if the reviewer would like that changed then we will be
happy to discuss.

Line 115: e.g. There are more works focus on MSGL.

Response: Other example references as suggested.

Line 165: Newton et al. A short description of the seafloor MSGL should be included in this work too.

Response: Agreed. Added as suggested.

Line 193: cannot not?

Response: Typo and corrected.

Line 196: (1 Ma) onwards. Add the age information.

Response: Agreed. Edited as suggested.

     Reviewer #2 – anonymous

It has been a pleasure to read this very well written manuscript. I am not a geophysicist, but I find the
presentation of the geophysical data and the interpretations of the data to be very logical and understandable.
The authors do a good job of outlining why this unusual record of multiple periods of ice streaming to the
palaeo shelf edge is important and what its implications are.

Response: We are grateful to the reviewer for their comments and have made all the required edits. We also
thank them for their quick turnaround of the review.

I have only one complaint. Given how unusual it is to have preservation of aggradational topset strata and given
the paucity of 3-D seismic data around Greenland (I presume it is not abundant), how do we know how
extensive ice streams were elsewhere? In the abstract, but not in the conclusions, you state: 'This suggests that
the ice streams that occupied Melville 21 Bugt during the Middle and Late Pleistocene were more active and
extensive than elsewhere in Greenland.' I think it is less presumptive and likely more correct to state that these
have not been observed elsewhere around Greenland. If other areas of the palaeo shelf had 3-D seismics and
aggradational topset strata, then perhaps one would find that Melville Bugt is not the only area where ice
advanced multiple times to the palaeo shelf edge. As you state, the record of past glacier ice extent is
fragmentary. You have found a very informative fragment and even better, you can correlate your unusual
preserved shelf strata to slope deposits (Knutz. Etal.,2019). I think it is more likely than not that many parts of
the Greenland shelf had extensive ice at the same times as shown by the Melville Bugt record. Modeling might
actually show what type of configuration of the ice sheet is likely and where/when more of these deposits
might be found, guided by your observations. I think the conclusions are well written and would be a good
model for revising the abstract. The unusual preservation of topset beds is interesting. Why is there is
accommodation space in Melville Bugt. You say it is because the ice streams are not occupying the whole
trough and so not eroding previous sediments. Did I understand that correctly. Is Melville Bugt unusually wide
compared to other troughs? It looks like it is wider than most.

Response: Fair point. This interpretation had been based on longer geological core records that have been
published from elsewhere on the margin, but our writing clearly needs a little refinement to make sure it is
clear what we mean. Abstract text has been modified as suggested and we have left this component of
comparisons for just the discussion section, as with the stated limitation on data availability it is an observation
that carries important caveats that is a discussion point, rather than something for the abstract. Regarding the
second point about the available accommodation, this is a good point about its width. The reviewer's
explanation is correct and we have added a short narrative that helps to make the thinking clearer on this point
as to the potential reasons why aggradational topsets have been preserved. This edit should clarify our current
understanding of why we think the aggradational topsets have been preserved.

A couple of minor comments:

P9 line 166...delete repeated ... and say multiple steaming events. 193...cannot not. Double negative or needs
editing?

Response: Rogue "not". Edits made as suggested.

199 and another example Line 241 and other places in the manuscript. Unattended this. Try to add what 'this' is
each time, just as you did in line 196, where you say 'this gradual shift'. It will make the writing even more clear.

Response: Great point on writing style, edited as suggested to improve clarity.

---

## Author Response (AR2)

Dear Pippa,

Thank you for the additional advice and comments below that have helped to tighten up the text. We have made changes in relation to every suggestion and these are outlined in the line responses below.

Best wishes,
Andrew Newton (on behalf of co-authors)

**Review of 'Repeated ice streaming on the northwest Greenland shelf since the onset of the Middle Pleistocene Transition' by Newton et al.**

I thank the authors for addressing the major scientific issues raised by the reviewers. I find the scientific content of the article to be robust, but a number of details need to be clarified before the article can be accepted for publication. These are outlined below, note that a couple relate to points that were not fully addressed from the original review.
Kind regards,
Pippa Whitehouse
Associate Editor, The Cryosphere

Points not addressed from initial review

1. Please check all instances of 'this' and 'these' – it is not always clear what you are referring to. For example, line 40: 'these studies' (a number of studies are mentioned, but only at the end of the sentence); line 50: 'these long-term changes'; line 180: 'these new data'.

Response: Once all the edits below were completed we have searched through the text for every "this/there" and changed a number of instances where "this/these" were used, including those listed. Some remain, but in each instance it is clear what is being referred to.

2. Use of brackets interrupts the text in a number of places. One example is the opening paragraph of section 2, but there are examples elsewhere.

Response: Whether they interrupt the text or not is a very subjective issue and difficult to judge what is classed as an interruption. In accordance with the original comments we changed some in the original manuscript as suggested when they appeared appropriate. In light of the editor highlighting this issue further, in the revised version the noted example has been changed, along with several others, to try and reduce the use of parenthesis as much as possible. This marginally adds to the word count. There is a major reduction in usage, but a number do remain as we believe these are actually helpful in reading these parts of the text – e.g. a reminder of dates of units.

Points in the main text
Line 19-20: 'ice streams continued to be active and extensive on the shelf during glacial stages' – statement is rather vague, largely because it is not clear when the ice streams were first active

Response: Edited the sentence to make the suggested timings clearer.

Line 20: here and elsewhere, please try to clarify that you are referring to the 'continental shelf' – many readers of The Cryosphere also think about ice shelves, so it is good to be clear

Response: Good point. In all instances where the word shelf is presented "continental" has been added where necessary to ensure clarity. This has also resulted in a slight tweak to the manuscript title. The only places where it has not been changed is in the term "palaeo-shelf/shelves" – we have looked at each sentence and believe it is clear what is meant by this term in each one.

Line 27: temperature is not the only driver of ice sheet change; the Knutz et al. (2019) article does not make any reference to temperature as a driver of change

Response: The Knutz et al. (2019) paper shows major ice sheet changes associated with glacial-interglacial stages without explicitly referring to temperatures changes but drawing links with interglacial transitions, and therefore increased temperatures as part of that picture, being responsible for changing ice flow characteristics in subsequent glacial stages. The wording here has been trimmed to just state the key observation from that paper, that the ice sheet massively changed in extent between several glacial-interglacial stages, without inferring causality.

Line 28: word missing – 'the future evolution…'

Response: Changed as suggested.

Line 30: the statement about needing to understand how the GrIS 'responded to warming' does not provide motivation for any of the research presented here, it is not relevant to this article

Response: This issue was raised in the previous review of the manuscript and we maintain it is a reasoned motivation to attempt to understand the glacial maximums because they provide information on the ice sheet characteristics at the start of any response to warming – i.e. the starting position for the retreat is part of the wider picture for understanding retreat. We think this is reasonable and it is actually discussed later in the text with regards to the roles of ice streams at the maximum extent in bringing about rapid ice sheet changes during a phase of warming (e.g. in the North Sea LGM paper by Sejrup et al. that is cited). However, we take the editor's advice and have replaced this text by mentioning the role of antecedent geology, which is more explicitly discussed.

"To better project the future evolution of the northwest Greenland ice sheet, and the GrIS as a whole, requires the reconstruction of past configurations of the ice sheet, the role and evolution through time of its ice streams, and an understanding of how the antecedent and evolving topography impacted ice flow patterns during past glacial stages."

Line 53: 'the mid and upper-slope' – of what?

Response: Changed to "middle and upper continental slope".

Line 57: 'a number of glacial advances' – statement is rather vague, see also lines 183-184, 271, 273

Response: In all the highlighted sections the text has been tweaked to improve clarity.

Line 76: suggest 'show the ice stream reached' -> 'show that fast-flowing ice reached' – in the original version it is ambiguous what ice stream you are referring to

Response: Edited as suggested with additional clarity to ensure the ice stream being referred to is clearer:

"The MSGL on the outermost continental shelf show that fast-flowing ice occupied the Melville Bugt Trough and reached the shelf edge, before retreating and experiencing changes in ice flow pathways, as is indicated by cross-cutting MSGL on the middle continental shelf (Newton et al., 2017)."

Line 83: suggest 'accumulation' -> 'sediment accumulation'

Response: Changed as suggested.

Line 100: 'seafloors' -> 'seafloor'

Response: Changed as suggested.

Line 111: 'the sets of MSGL' – the landforms on the buried palaeo-seafloor surfaces are not yet identified as MSGL,
this methods section just talks about identifying 'glacial landforms'

Response: Good point. Changed to remove reference to MSGL.

Line 131: perhaps quote the bearing of MSGL set 1, given that you do for all other sets

Response: Added as suggested.

Line 137: 'The MSGL…' – make it clear you are still talking about MSGL set 2

Response: Edited as suggested.

Line 138-139: 'MSGL sets 3 and 4 lie in the topset strata of unit A9' – clarify that MSGL sets 3 and 4 are located
on separate surfaces, i.e. that they reflect separate glacial advances

Response: Edited as suggested.

Lines 153-154: the seismic profile in which MSGL set 5 is identified is orientated very close to the direction of the
mapped MSGLs (fig. 6); perhaps comment on how this influences your ability to identify MSGL set 5 in the 2D
data

Response: short sentence added as suggested.

Line 159: include a reference to figure 6 to help the reader identify the location of MSGL set 6

Response: Changed as suggested.

Lines 177-178: line 121 states that buried MSGLs have been observed on other margins. If some of these buried
MSGLs are thought to pre-date the LGM then it would seem appropriate to clarify that the statement on line 177-
178 specifically relates to the Greenland margin

Response: Good point, changed as suggested.

Line 191-192: 'changes in depocentre migration and MSGL orientation, such as presented here, may have forced
modifications in ice sheet flow…' – statement does not really make sense, how can a change in MSGL orientation
force a modification in ice sheet flow? The later part of the sentence talks about accommodation space, but the
logic of the early part is muddled.

Response: Agreed. Sentence has been significantly reworded to:
"The reasons for this change are unresolved, but modification of the submarine topography brought about by
glacigenic deposition and erosion, such as presented here, may have forced adjustments in the ice sheet flow on
the outer continental shelf due to changes in available accommodation."

Line 201-202: refer to figure 5b?

Response: Changed as suggested (though to fig 7 due to figure suggestion).

Line 206: refer to figure 6
Response: Changed as suggested.
Line 218: 'This northward' -> 'The northward'
Response: Changed as suggested.
Line 221: check use of 'subsequent' here, replace with 'successive' ?
Response: Changed as suggested.
Line 225: 'the Melville Bugt Ridge' – please label this feature on a figure
Response: This is labelled on Fig. 2. The in-text citations provide a source for anybody wishing to read up on this
inversion structure further.
Line 227-228: please provide a brief explanation (or a reference) to support the statement that an increase in
water depth would reduce erosion of the topsets
Response: Text added "… water depths to the point where ice grounding was reduced or removed".
Line 242: 'IRD' – please define acronym
Response: Removed IRD and added the full title as IRD is no longer discussed elsewhere.
Line 246: suggest 'As' -> 'While' or 'Although'
Response: Changed as suggested.
Lines 259-262: I could not find any evidence in Willeit et al. (2019) to support the statement 'recent modelling …
(Willeit et al., 2019) suggests multiple ice sheet reconstructions that do not capture the ice sheet extent that has
been inferred from buried landform records on many glaciated margins (e.g. Rea et al., 2018), including Melville
Bugt.' Please justify this statement in your rebuttal. There is no need to edit the manuscript, but I am keen to
check that this criticism of previous work is robust.
Response: The Willeit et al paper does not make that statement specifically, the text is meant to refer to the
observation that ice sheet extent derived from their models does not match up to those published and presented
in this paper or in Rea et al. This is our own observation and a slight rewording has been carried out to make this
distinction clearer.
"For example, recent modelling exploring Pleistocene climate evolution (Willeit et al., 2019) provides palaeo-
geographic maps of ice sheet extent that do not capture the ice sheet extent inferred from buried landform
records on many glaciated margins (e.g. Rea et al., 2018), including Melville Bugt. Thus, there is currently a
mismatch between modelling outputs and landform records."
Line 270: 'anywhere' does not make sense in its current position; edit or delete
Response: Deleted as suggested.
Figures
Figure 2 caption states that units A7-A9 likely cover the middle Pleistocene (781-126 ka) and the transition into it
at 1.3 Ma, but this disagrees with information on lines 131, 133, 139 and figure 6d

Response: Text has been refined to ensure dating is clearer.

"The tentative chronology from Knutz et al. (2019) suggests that the palaeo-seafloor surfaces preserved within units A7-A9 likely cover a time period from ~1.3-0.43 Ma. This captures much of the Middle Pleistocene (781-126 ka) and the transition into it from ~1.3 Ma."

Figure 3b: what is the purpose of the white dashes next to the unit labels? Do the red lines identify the upper or lower boundary of each sediment unit? Caption to this figure refers to a cross-section and a profile – try to use consistent terminology

Response: The white dash is used to help clearly indicate which label is attributed to each surface due to the condensed nature of the stratigraphy. Each surface is the upper boundary of that unit. Caption modified to improve clarity.

Figure 4 caption: 'Orange arrows…' – make it clear you are now talking about panels (d) to (f). Please identify which set of MSGL are shown in panels (d) to (f) and in which unit each of these palaeo-surfaces is located (reference to figure 6 may be useful). Please explicitly state what the contoured shapes represent in panels (a) to (c).

Response: The figure has been edited with extra labels. Caption has been fully rewritten:
"Buried MSGL and associated TMF thickness maps. Panels (a) to (c) show the geographic location of MSGL sets 2-4 displayed as hillshade images on panels (d) to (f). The dashed grey line on (a) to (c) is the 3D seismic survey outline overlain on the contemporary seafloor, the white arrows show the inferred ice flow direction from the MSGL, and the contoured outlines show the thickness of the sedimentary deposit associated with each MSGL set. Orange arrows on panels (d) to (f) show the inferred ice flow direction. On panel (d) the green line displays the location of the inset cross-section profile of the MSGL. Blue arrows point to the mounded features visible on the hillshade image. The red circles in (d) to (f) display average MSGL compass bearings (black line) and the standard deviation (blue fan beneath) for each panel. Location of panels (a) to (c) shown on Fig. 1. The relative ages and stratigraphic positons of each MSGL set are discussed in the text and labelled on Fig. 6."

Figure 5b does not seem to be related to figure 5a. It is most closely related to the text on ice stream migration in section 5; suggest separating figure 5b and moving it to after figure 6.

Response: Changed as suggested.

Figure 6 caption: make it clear that the 'LGM record' is the same as MSGL set 6. Note that there is no compass bearing for MSGL set 5.
In general, use of outlined text in figures is difficult to read, especially for smaller figures. Please try to improve image quality where necessary.

Response: Caption edited as suggested.

Regarding the text with halos, the figures were originally created without the halo but it was felt that with the desire to provide multiple colours to make it easier to cross-reference between different panels the halo was necessary. The size of the text has been increased to improve visibility. At 100% scale the text should be easily visible.

[revised manuscript text omitted]

---

## Editor Decision (ED2)

**Review of 'Repeated ice streaming on the northwest Greenland shelf since the onset of the Middle Pleistocene Transition' by Newton et al.**

I thank the authors for addressing the major scientific issues raised by the reviewers. I find the scientific content of the article to be robust, but a number of details need to be clarified before the article can be accepted for publication. These are outlined below, note that a couple relate to points that were not fully addressed from the original review.

Kind regards,

Pippa Whitehouse
Associate Editor, The Cryosphere

Points not addressed from initial review

1. Please check all instances of 'this' and 'these' – it is not always clear what you are referring to. For example, line 40: 'these studies' (a number of studies are mentioned, but only at the end of the sentence); line 50: 'these long-term changes'; line 180: 'these new data'.

2. Use of brackets interrupts the text in a number of places. One example is the opening paragraph of section 2, but there are examples elsewhere.

Points in the main text

Line 19-20: 'ice streams continued to be active and extensive on the shelf during glacial stages' – statement is rather vague, largely because it is not clear when the ice streams were first active

Line 20: here and elsewhere, please try to clarify that you are referring to the 'continental shelf' – many readers of The Cryosphere also think about ice shelves, so it is good to be clear

Line 27: temperature is not the only driver of ice sheet change; the Knutz et al. (2019) article does not make any reference to temperature as a driver of change

Line 28: word missing – 'the future evolution…'

Line 30: the statement about needing to understand how the GrIS 'responded to warming' does not provide motivation for any of the research presented here, it is not relevant to this article

Line 53: 'the mid and upper-slope' – of what?

Line 57: 'a number of glacial advances' – statement is rather vague, see also lines 183-184, 271, 273

Line 76: suggest 'show the ice stream reached' -> 'show that fast-flowing ice reached' – in the original version it is ambiguous what ice stream you are referring to

Line 83: suggest 'accumulation' -> 'sediment accumulation'

Line 100: 'seafloors' -> 'seafloor'

Line 111: 'the sets of MSGL' – the landforms on the buried palaeo-seafloor surfaces are not yet identified as MSGL, this methods section just talks about identifying 'glacial landforms'

Line 131: perhaps quote the bearing of MSGL set 1, given that you do for all other sets

Line 137: 'The MSGL…' – make it clear you are still talking about MSGL set 2

Line 138-139: 'MSGL sets 3 and 4 lie in the topset strata of unit A9' – clarify that MSGL sets 3 and 4 are located on separate surfaces, i.e. that they reflect separate glacial advances

Lines 153-154: the seismic profile in which MSGL set 5 is identified is orientated very close to the direction of the mapped MSGLs (fig. 6); perhaps comment on how this influences your ability to identify MSGL set 5 in the 2D data

Line 159: include a reference to figure 6 to help the reader identify the location of MSGL set 6

Lines 177-178: line 121 states that buried MSGLs have been observed on other margins. If some of these buried MSGLs are thought to pre-date the LGM then it would seem appropriate to clarify that the statement on line 177-178 specifically relates to the Greenland margin

Line 191-192: 'changes in depocentre migration and MSGL orientation, such as presented here, may have forced modifications in ice sheet flow…' – statement does not really make sense, how can a change in MSGL orientation force a modification in ice sheet flow? The later part of the sentence talks about accommodation space, but the logic of the early part is muddled.

Line 201-202: refer to figure 5b?

Line 206: refer to figure 6

Line 218: 'This northward' -> 'The northward'

Line 221: check use of 'subsequent' here, replace with 'successive' ?

Line 225: 'the Melville Bugt Ridge' – please label this feature on a figure

Line 227-228: please provide a brief explanation (or a reference) to support the statement that an increase in water depth would reduce erosion of the topsets

Line 242: 'IRD' – please define acronym

Line 246: suggest 'As' -> 'While' or 'Although'

Lines 259-262: I could not find any evidence in Willeit et al. (2019) to support the statement 'recent modelling … (Willeit et al., 2019) suggests multiple ice sheet reconstructions that do not capture the ice sheet extent that has been inferred from buried landform records on many glaciated margins (e.g. Rea et al., 2018), including Melville Bugt.' Please justify this statement in your rebuttal. There is no need to edit the manuscript, but I am keen to check that this criticism of previous work is robust.

Line 270: 'anywhere' does not make sense in its current position; edit or delete

Figures

Figure 2 caption states that units A7-A9 likely cover the middle Pleistocene (781-126 ka) and the transition into it at 1.3 Ma, but this disagrees with information on lines 131, 133, 139 and figure 6d

Figure 3b: what is the purpose of the white dashes next to the unit labels? Do the red lines identify the upper or lower boundary of each sediment unit? Caption to this figure refers to a cross-section and a profile – try to use consistent terminology

Figure 4 caption: 'Orange arrows…' – make it clear you are now talking about panels (d) to (f). Please identify which set of MSGL are shown in panels (d) to (f) and in which unit each of these palaeo-surfaces is located (reference to figure 6 may be useful). Please explicitly state what the contoured shapes represent in panels (a) to (c).

Figure 5b does not seem to be related to figure 5a. It is most closely related to the text on ice stream migration in section 5; suggest separating figure 5b and moving it to after figure 6.

Figure 6 caption: make it clear that the 'LGM record' is the same as MSGL set 6. Note that there is no compass bearing for MSGL set 5.

In general, use of outlined text in figures is difficult to read, especially for smaller figures. Please try to improve image quality where necessary.

---

## Author Response (AR3)

Dear authors,

Thank you for promptly addressing all the points raised during the review process. This is a novel study that
clearly documents evidence for multiple episodes of ice streaming across the continental shelf of northwest
Greenland. As such, the findings of this study will be a useful target for subsequent modelling efforts.

In implementing the edits, a couple of minor errors have crept into the manuscript. These are documented
below and should be resolved as you prepare the final version of the manuscript, but otherwise I am happy to
accept this article for publication in The Cryosphere.

Pippa Whitehouse Associate Editor, The Cryosphere

Response: We are thankful for the editor's support of this work and the meticulous review that spotted some
rogue minor errors. All have been modified as suggested and we thank the editor for their handling and
editorial perseverance on this manuscript.

Minor errors – line numbers refer to manuscript version 4

Line 68: delete one instance of 'that'

Response: Corrected.

Line 102: 'two-way-travel time' (as in the caption to figure 2) Line 137: 'their' -> 'the MSGL'

Response: Edited as suggested.

Line 138: (Fig. 4a, d) – to bring in line with text on line 145

Response: Edited as suggested.

Line 149: 'displayed' -> 'which are displayed'

Response: Edited as suggested.

Line 163: Text is a little ambiguous following insertion of the extra sentence. Suggest "The interpretation of the
corrugated features as MSGL set 5 …"

Response: Edited as suggested.

Line 168: "The final set of MSGL… has been interpreted as a grounded ice stream" – suggest editing to indicate
that the MSGL were produced by/provide evidence for a grounded ice stream

Response: Edited as suggested to: "The final set of MSGL (set 6) is observed in unit A11 (~0.35-0 Ma) on the
seafloor and provides evidence for a grounded ice stream on the outer continental shelf at the LGM (Newton et
al., 2017) (Fig. 6c)."

Line 182: Could flag up your contribution – "…means that prior to this study ice stream…"

Response: Edited as suggested.

Line 185: Word missing – "Using the new seismic data, …"

Response: Edited as suggested.

Line 243: 'each MSGL surfaces' – singular/plural issue

[revised manuscript text omitted]

---

## Editor Decision (ED3)

Dear authors,

Thank you for promptly addressing all the points raised during the review process. This is a novel study that clearly documents evidence for multiple episodes of ice streaming across the continental shelf of northwest Greenland. As such, the findings of this study will be a useful target for subsequent modelling efforts.

In implementing the edits, a couple of minor errors have crept into the manuscript. These are documented below and should be resolved as you prepare the final version of the manuscript, but otherwise I am happy to accept this article for publication in The Cryosphere.

Pippa Whitehouse
Associate Editor, The Cryosphere

Minor errors – line numbers refer to manuscript version 4

Line 68: delete one instance of 'that'

Line 102: 'two-way-travel time' (as in the caption to figure 2)

Line 137: 'their' -> 'the MSGL'

Line 138: (Fig. 4a, d) – to bring in line with text on line 145

Line 149: 'displayed' -> 'which are displayed'

Line 163: Text is a little ambiguous following insertion of the extra sentence. Suggest "The interpretation of the corrugated features as MSGL set 5 …"

Line 168: "The final set of MSGL… has been interpreted as a grounded ice stream" – suggest editing to indicate that the MSGL were produced by/provide evidence for a grounded ice stream

Line 182: Could flag up your contribution – "…means that prior to this study ice stream…"

Line 185: Word missing – "Using the new seismic data, …"

Line 243: 'each MSGL surfaces' – singular/plural issue